# Identification of Novel Antiviral Compounds Targeting Entry of Hantaviruses

**DOI:** 10.3390/v13040685

**Published:** 2021-04-16

**Authors:** Jennifer Mayor, Giulia Torriani, Olivier Engler, Sylvia Rothenberger

**Affiliations:** 1Institute of Microbiology, University Hospital Center and University of Lausanne, Rue du Bugnon 48, CH-1011 Lausanne, Switzerland; jennifer.mayor@unil.ch (J.M.); giulia.torriani@supsi.ch (G.T.); 2Spiez Laboratory, Swiss Federal Institute for NBC-Protection, CH-3700 Spiez, Switzerland; Olivier.Engler@babs.admin.ch

**Keywords:** orthohantaviruses, arenavirus, ebolavirus, viral hemorrhagic fever, antiviral screening, entry inhibitors, GFP

## Abstract

Hemorrhagic fever viruses, among them orthohantaviruses, arenaviruses and filoviruses, are responsible for some of the most severe human diseases and represent a serious challenge for public health. The current limited therapeutic options and available vaccines make the development of novel efficacious antiviral agents an urgent need. Inhibiting viral attachment and entry is a promising strategy for the development of new treatments and to prevent all subsequent steps in virus infection. Here, we developed a fluorescence-based screening assay for the identification of new antivirals against hemorrhagic fever virus entry. We screened a phytochemical library containing 320 natural compounds using a validated VSV pseudotype platform bearing the glycoprotein of the virus of interest and encoding enhanced green fluorescent protein (EGFP). EGFP expression allows the quantitative detection of infection and the identification of compounds affecting viral entry. We identified several hits against four pseudoviruses for the orthohantaviruses Hantaan (HTNV) and Andes (ANDV), the filovirus Ebola (EBOV) and the arenavirus Lassa (LASV). Two selected inhibitors, emetine dihydrochloride and tetrandrine, were validated with infectious pathogenic HTNV in a BSL-3 laboratory. This study provides potential therapeutics against emerging virus infection, and highlights the importance of drug repurposing.

## 1. Introduction

Orthohantaviruses (hereafter referred to as hantaviruses), in the family Hantaviridae, are human pathogenic viruses causing severe diseases with high mortality rate and represent therefore a serious challenge for public health. The prototypic Hantaan (HTNV) and Seoul (SEOV) viruses are widespread in Asia, where they cause hemorrhagic fever with renal syndrome (HFRS), with up to 15% case-fatality. In the Americas, the Sin Nombre (SNV) and Andes (ANDV) viruses are associated with hantavirus cardiopulmonary syndrome (HCPS), with up to 40% mortality [1,2]. Orthohantaviruses are distributed worldwide, which corresponds to the geographical distribution of their reservoir hosts, namely rodents, shrews, moles and bats.

Human infections are accidental and occur through inhalation of contaminated rodent excreta. Transmission via aerosols requires the capacity to break the species barrier and productively infect cells of the human respiratory epithelium. Upon first replication at the site of entry, orthohantaviruses disseminate systemically through the bloodstream. The therapeutic options are currently limited to supportive care and the only available vaccine, Hantavax^®^, is approved in several Asian countries, but not in Europe or the U.S.A. [3,4]. Given the worldwide distribution of orthohantaviruses, novel medical approaches for treatment and prevention are of high priority.

Orthohantaviruses, arenaviruses and filoviruses can cause viral hemorrhagic fevers (VHFs) and their life cycle shows some common features, especially at the level of viral attachment and entry [5,6,7]. Viral attachment and entry represent the first and most fundamental steps in virus zoonotic transmission and infection. It therefore represents a promising target for therapeutic intervention in order to block viruses before they can take control of the host cell and produce viral progeny. In this study, using a fluorescence-based screening assay, we identified new antivirals against HTNV and ANDV, as well as Lassa virus (LASV) and Ebola virus (EBOV). The screening assay was performed using a validated pseudotype platform based on recombinant vesicular stomatitis reporter virus (VSV) bearing the glycoprotein of interest and encoding enhanced green fluorescent protein (EGFP) [8,9,10]. EGFP expression allows the quantitative detection of infection and the identification of compounds affecting viral entry.

As 80% of the world population depends on traditional medicine [11], including the use of herbal medicinal products, we screened a phytochemical library containing 320 natural compounds and identified novel small molecular compounds with antiviral activity against HTNV and ANDV, as well as LASV and EBOV. Together, our data demonstrate that our screening platform can detect compounds with varying degrees of antiviral activity and therefore facilitates drug discovery to fight emerging virus infections.

## 2. Materials and Methods

### 2.1. Cells

Human lung carcinoma alveolar epithelial (A549) cells were maintained in Dulbecco’s modified Eagle medium (DMEM)-10% [*v*/*v*] fetal calf serum (FCS) at 37 °C under 5% CO_2_ atmosphere. Monkey kidney epithelial (Vero E6) cells were maintained in Biochrom minimum essential media (MEM) with Earle’s salts supplemented with 10% [*v*/*v*] FCS, 0.625% l-glutamine, 0.5% penicillin-streptomycin and 0.5% NEAA (Biochrom, Cambridge, United Kingdom) at 37 °C. For infection studies, cells were seeded in 24-, 48- or 96-well plates and cultured 16 h until cell monolayer formation.

### 2.2. Plasmids and Reagents

pWRG/HTNV-M [12] was kindly provided by Connie. S. Schmaljohn (U.S. Army Medical Research Institute of Infectious Diseases, Fort Detrick, MD, USA). The expression plasmid pI18 for the GPC of ANDV strain CHI-7913 was kindly provided by Nicole Tischler (Molecular Virology Laboratory, Fundación Ciencia & Vida, Santiago, Chile) and has been described previously [13]. Expression plasmids encoding Lassa virus GPC strain Josiah and VSV-G were reported previously [8]. The expression vector encoding Ebola virus strain Makona was kindly provided by Mark Page (The National Institute for Biological Standards and Control, South Mimms, UK).

The phytochemical library was from Prestwick Chemicals (stock concentration 10 mM in DMSO, Illkrich Graffenstaden, France). The selected positive hits monensin sodium salt, rotenone, tetrandrine, as well as 5-(*N*-ethyl-*N*-isopropyl) amiloride (EIPA) were from Sigma-Aldrich (Buchs, Switzerland), nitrarine dihydrochloride was provided by Latoxan (Portes lès Valence, France), and emetine dihydrochloride was from EMD Millipore (Darmstadt, Germany). The CellTiter-Glo^®^ Assay System was obtained from Promega (Madison, WI, USA).

### 2.3. Pseudotype Virus Production and Viruses

The pseudotype viral system is based on the recombinant VSV*ΔG-Luc vector in which the glycoprotein gene (G) had been deleted and replaced with genes encoding green fluorescent protein (EGFP; indicated by an asterisk) and luciferase (Luc) [14]. Pseudoviruses were generated as reported previously [8,9]. Recombinant human adenovirus serotype 5 (AdV5) expressing EGFP has been described previously [15]. HTNV strain 76/118 [16] was propagated in Vero E6 cells (Vero C1008, ATCC CLR 1586) in the BSL-3 containment laboratory at Spiez Laboratory.

### 2.4. Microscopy

ImageXpress^®^ Micro XLS Widefield High-Content Screening System (Molecular Devices, San Jose, CA, USA) was used to visualize and quantify the number of EGFP-infected cells. Per each well, one image, covering 5.32% of the well surface, was captured using two fluorescent channels, i.e., DAPI and GFP. Images were then analyzed with MetaXpress software. To detect and analyze EGFP-infected cells, two settings were fixed: the typical cell area and the GFP expression. Cell area was first determined by the presence of nuclei based on DAPI stained images, and with several manual delimitations of cell edges based on bright field images. Upon first identification, the number of cells expressing EGFP was determined.

### 2.5. Dynamic Range Determination

The titration of VSV*ΔG-Luc(VSV-G) was performed by two-fold serial dilution on monolayers of Vero E6 cells. After 1 h of infection, fresh medium was added, and cells were incubated overnight at 37 °C. Infection was quantified by measuring EGFP-positive infected cells per well site using ImageXpress^®^ Micro XLS Widefield High-Content Screening System (Molecular Devices). Data were analyzed with MetaXpress and GraphPad Prism 7 softwares.

### 2.6. ROC Curve

Confluent monolayers of A549 cells were either treated with 50 μM EIPA for 30 min (25 wells) or untreated (25 wells), followed by infection with VSV*ΔG-Luc(HTNV-G) in the presence of the inhibitor. After 16 h of incubation, cells were washed with medium, fixed with 2% PFA, and the nuclei were stained with NucBlue^TM^ Fixed Cell ReadyProbes^TM^ Reagent (DAPI) following manufacturer’s instruction (ThermoFisher, Waltham, MA, United States). Data were acquired and analyzed with MetaXpress and GraphPad Prism 7 softwares.

### 2.7. Antiviral Screening

Confluent monolayers of A549 cells were pre-treated with candidate compounds dissolved in DMEM 10% FCS (final concentration 10 μM) for 45 min. Around 2.5 × 10^5^ infectious unit (IU)/mL (MOI of 0.3) of VSV*ΔG-Luc(HTNV-G), VSV*ΔG-Luc(ANDV-G), VSV*ΔG-Luc(EBOV-G), or VSV*ΔG-Luc(LASV-G) were added in the presence of candidate compounds. After 2 h of infection with indicated pseudoviruses, the inoculum was removed, and the cells were washed and incubated for 16 h in complete medium containing 20 mM ammonium chloride. Infection was first quantified by measuring EGFP-positive infected cells by fluorescent microscopy, and analyzed with MetaXpress software. Compounds demonstrating ≥ 20% reduction in cellular ATP levels according to CellTiter-Glo^®^ assay (Promega) were removed from the screen.

### 2.8. Drug Inhibition Assay

Confluent monolayers of A549 or Vero E6 cells were pre-treated with drugs for 30 min at 37 °C under 5% CO_2_, followed by infection with the indicated viruses in presence of the drugs for 1.5 h at 37 °C. After incubation, cells were washed twice with DMEM 10% FCS supplemented with 20 mM of ammonium chloride and incubated for 16 h at 37 °C with the presence of ammonium chloride. Finally, infection was quantified by counting EGFP-positive infected cells per well using an EVOS FLoid Cell Imaging Station (ThermoFisher).

For time-of-addition experiments, infection with the indicated viruses was performed in A549 cells. Then, 1.5 h post-infection, the cells were subsequently washed twice with DMEM 10% FCS supplemented with 20 mM of ammonium chloride and incubated for 16 h at 37 °C with the presence of ammonium chloride and appropriate concentration of each antiviral. Productive infection was quantified by counting EGFP-positive infected cells per well using an EVOS FLoid Cell Imaging Station (ThermoFisher).

### 2.9. Pathogenic Virus Validation

Vero E6 cells were pre-treated with increasing concentrations of EIPA, monensin sodium salt, emetine dihydrochloride and tetrandrine for 30–45 min at 37 °C, followed by HTNV (strain 76/118) infection for 1 h at 37 °C in presence of the drug. Cells were subsequently washed once with MEM 2% FCS and incubated with MEM 2% FCS containing appropriate drug concentrations for 2 days.

To calculate the therapeutic index (TI), confluent monolayers of Vero E6 cells were treated with EIPA, tetrandrine, and emetine dihydrochloride starting at 100 µM concentration, followed by a ~1.5-fold dilution for two days. After 48 h, cells were treated with 1% triton-100 (diluted in water) for 15 min at 37 °C in order to totally lyse the cells and obtain the background values. Cell viability was then measured using CellTiter-Glo^®^ Assay System (Promega). The median toxic dose (TD_50_) was calculated based on the cell viability values, while the median effective dose (ED_50_) on cell infectivity with HTNV. Therapeutic index was finally calculated as follows: TD_50_/ED_50_.

### 2.10. RT-qPCR Analysis

Viral RNA was isolated using RNeasy^®^ Plus Universal Mini Kit (QIAGEN, Hilden, Germany) following the manufacturer’s instructions. Quantification of viral RNA was performed using a real-time quantitative polymerase chain reaction (RT-qPCR) assay specific for the HTNV nucleocapsid coding region in a LigthCycler^®^ 96 (Roche, Mannheim, Germany) following the manufacturer’s instructions. Cells treated in parallel with the drugs were analyzed for viability using the CellTiter-Glo^®^ Assay System (Promega).

### 2.11. Statistical Analysis

Graphical representation and statistical analysis were performed using GraphPad Prism 7 software. Data are means + SD (*n* = 3), and *p* values of <0.05 were considered statistically significant.

## 3. Results

### 3.1. Determination of the Dynamic Range for Luminescence Detection

To overcome the biosecurity limitation for work with highly pathogenic viruses, we used a validated pseudotype platform. Briefly, we used a recombinant vesicular stomatitis virus (VSV), in which the glycoprotein gene was deleted and replaced with reporter genes encoding enhanced green fluorescent protein (EGFP, indicated by an asterisk) and firefly luciferase (Luc). Then, glycoproteins of interest from HTNV, ANDV, EBOV or LASV were provided in *trans* in the recombinant VSV [5,8,9]. The VSV-pseudotypes systems are replication-competent, but propagation-deficient, making them suitable for work under BSL-2 conditions.

Recombinant viruses expressing a fluorescent reporter are valuable for the rapid identification of candidate antiviral compounds [17,18]. To determine whether GFP encoded in our pseudoviruses could be used as a surrogate readout to rapidly test the activity of antiviral drug candidates, we first measured the number of EGFP-positive infected cells. Confluent monolayers of A549 cells were infected with two-fold serial dilution of VSV*ΔG-Luc(VSV-G) for 16 h. The cells were then fixed with 2% PFA for 20 min at RT and the nuclei were stained. The fluorescent microscope ImageXpress^®^ Micro XLS Widefield High-Content Screening System (Molecular Devices) was used to measure and analyze the number of infected cells according to the average size of infected cells expressing EGFP (Figure 1A,B). As expected, the EGFP signal could be detected, when the viral concentration is between 5 × 10^7^ and 2 × 10^5^ infectious units (IU)/mL, allowing the quantification of infected cells. Our results demonstrate that GFP expression represents a valuable tool to monitor the number of infected cells, and thus pseudovirus entry in our screening platform.

### 3.2. Establishment of a ROC Curve

To further determine whether GFP expression and the size-based method could be used to statistically evaluate the antiviral activity of selected compounds, we performed receiver-operating characteristics (ROC) analysis. An ROC curve is a test of probability to determine how well a screening assay distinguishes treated infected cells and non-treated infected cells. More specifically, it shows the trade-off between sensitivity and specificity. Sensitivity is the proportion of treated cells identified correctly when a treatment is present. Similarly, specificity is the proportion of non-treated cells when no treatment is added. In other words, it is the probability to exclude negative results. An ideal ROC curve is a vertical line on the *y*-axis (specificity = 1.0) with a horizontal line at sensitivity = 1.0, while for an assay that is not better than random prediction, it is a diagonal line. The area under the curve (AUC) gives information about the model performances. An ideal AUC is 1.0, whereas a value of 0.5 corresponds to a random assay.

As many emerging human pathogenic viruses are transmitted via aerosols, we used A549 human lung epithelial cells as the model [1,2,19], which are also known to be susceptible to HTNV infection [20]. To test the screening platform and confirm the potential to find novel antiviral compounds, we used VSV*ΔG-Luc(HTNV-G) and 5-(*N*-ethyl-*N*-isopropyl)-amiloride (EIPA) previously shown to inhibit HTNV-G pseudovirus entry [9]. A549 cells were either non-treated or treated with 50 μM EIPA for 30 min, followed by infection with VSV*ΔG-Luc(HTNV-G) in the presence of the inhibitor. After 16 h of incubation, cells were washed, fixed with 2% PFA, and stained. The number of infected cells was then analyzed through GFP expression and the size-based method using a fluorescent microscope (Figure 2A).

We obtained an AUC of 1 to discriminate between infected and non-infected cells (Figure 2B), and an AUC of 0.9483 between treated and non-treated infected cells (Figure 2C). This means that infected cells treated with EIPA have a 94.8% chance of being discriminated from non-treated infected cells. These results demonstrate the feasibility of using GFP expression to assess the antiviral activity of selected compounds.

A549 human lung epithelial cells are a suitable cell model to study aerosol-transmitted viruses. However, HTNV, ANDV, and EBOV are generally propagated on Vero E6 cells and the HTNV strain 76–118 is adapted to this cell line [16,21]. Using HTNV-G pseudotypes, we thus examined whether Vero E6 constitute a cell model to test the impact of EIPA. Confluent monolayers of Vero E6 or A549 cells were pre-treated with increasing concentrations of EIPA for 45 min, and subsequently infected with VSV*ΔG-Luc(HTNV-G). After 90 min, cells were washed and complete medium with 20 mM ammonium chloride was added to cells in order to block further entry via low pH-triggered membrane fusion. VSV*ΔG-Luc(HTNV-G) is similarly inhibited in both Vero E6 and A549 cells, demonstrating an inhibition of ~78% and 70% at 30 µM, respectively (Figure 2D).

Moreover, the inhibitory activity of EIPA was also evaluated on pathogenic HTNV infection. Confluent monolayers of Vero E6 cells were incubated with various concentration of EIPA for 45 min, followed by HTNV infection at 37 °C for 1 h. Cells were washed once and incubated with medium containing appropriate EIPA concentration for 48 h. We observed a decrease of ~65% in viral production at an EIPA concentration of 30 µM (Figure 2E). This suggests that entry inhibitors could be potent antiviral drugs, and provides a first validation of our screening platform.

### 3.3. High-Throughput Screening Assay for Identifying Entry Inhibitors

Plants can be a very valuable source of new antiviral molecules. To identify novel antiviral therapeutics against hantaviruses, we screened 320 natural compounds of the Prestwick Phytochemical library. Monolayers of A549 cells were pre-treated with 10 μM of compounds for 45 min and infected with VSV*ΔG-Luc(HTNV-G), VSV*ΔG-Luc(ANDV-G), VSV*ΔG-Luc(EBOV-G), or VSV*ΔG-Luc(LASV-G) in the presence of drugs. After 2 h, the drugs were washed out to minimize the duration of drug exposure and unwanted off-target effects. The cells were then incubated in medium supplemented with ammonium chloride to block further entry via low pH-triggered membrane fusion. The following day, cells were fixed with 2% PFA for 20 min, and the nuclei were stained (DAPI). Productive infection was detected by EGFP expression measurement with ImageXpress^®^ Micro XLS Widefield High-Content Screening System. To avoid artefacts due to toxicity, the candidate compounds underwent previous evaluation in a cell viability test that detects changes in cellular ATP levels under the exact assay conditions. Candidate inhibitors that resulted in >20% reduced cell viability were excluded from the screen. We found that 88% of tested compounds are not toxic (Figure 3A).

The screening was performed with HTNV-G and ANDV-G, as well as with EBOV-G and LASV-G in order to compare the effects of inhibitors on different viral families and discern between drugs with broad-spectrum effects or virus-specific effects. Drugs with broad-spectrum effects can be of value because novel emerging viruses have huge impact on affected populations. In our analysis, compounds inhibiting the infection >50% of at least one pseudovirus were considered as hits. We identified five compounds (~1.5% of the library) that reduced viral entry by >50%, namely tetrandrine, nitrarine dihydrochloride, rotenone, monensin sodium salt and emetine dihydrochloride (Figure 3B–E, highlighted in green). Tetrandrine and nitrarine dihydrochloride inhibited infection of EBOV-G, rotenone reduced ANDV-G infection and monensin sodium salt demonstrated an antiviral effect on HTNV-G. Emetine dihydrochloride showed a broad-spectrum inhibitory effect on HTNV-G, EBOV-G and LASV-G. Moreover, the hits identified did not demonstrate cytotoxicity. Overall, these results suggest that the screening platform we developed can detect the antiviral activity of small molecule compounds.

### 3.4. Characterization of Hits Identified in Primary Screen

To further characterize the hits and determine the antiviral activity of each drug, we mapped dose–response curves while titrating the compounds. Confluent monolayers of A549 cells were treated with increasing concentration of tetrandrine, nitrarine dihydrochloride, rotenone, monensin sodium salt or emetine dihydrochloride for 45 min at 37 °C, followed by incubation with the pseudoviruses in the presence of the drugs. The inoculum was then removed and the cells were further incubated in medium containing ammonium chloride to block secondary infection with low pH-triggered membrane fusion. Tetrandrine, nitrarine dihydrochloride and rotenone tended to have virus-specific antiviral activity (Figure 4A–C). The three compounds did not show any cytotoxicity under the assay conditions. We found that tetrandrine drastically decreased entry of VSV*ΔG-Luc(ANDV-G), VSV*ΔG-Luc(EBOV-G), and VSV*ΔG-Luc(LASV-G) and had a slightly lower antiviral activity against VSV*ΔG-Luc(HTNV-G). Nitrarine dihydrochloride demonstrated a similar reduction in infection for VSV*ΔG-Luc(EBOV-G), while VSV*ΔG-Luc(ANDV-G) entry was reduced at 85% at 7 µM concentration. Rotenone had antiviral activity mainly against VSV*ΔG-Luc(ANDV-G) and VSV*ΔG-Luc(LASV-G). All pseudoviruses tested were inhibited by both monensin sodium salt (Figure 4D) and emetine dihydrochloride, which demonstrated a reduction in infection in a dose-dependent manner (Figure 4E).

To delineate the impact of the compounds on viral entry and post-entry steps of infection, we performed a “time-of-addition” experiment. To do this, the inhibitors were added at specific concentrations at 1.5 h post-infection (Figure 5A). As a control, we used a recombinant adenovirus 5 (AdV5), a non-enveloped virus, that does not pass through the endosome to get into the cells. Tetrandrine, rotenone and monensin sodium salt reduced viral entry of HTNV-G, while the drug did not show antiviral activity against AdV5 (Figure 5B). Moreover, they did not demonstrate any antiviral activity at a post-entry step. These data showed that these compounds can affect the entry of tested VHFs, most likely by inhibiting their exit from the endosomal compartment.

In contrast, emetine dihydrochloride demonstrated a potent antiviral activity against AdV5 at entry level, indicating that the compound had an impact on later steps of the viral cycle. Since AdV5 at the post-entry level was also completely inhibited by emetine dihydrochloride, this suggests that the compound had no effect on the endosomes, but most probably on other steps of the viral infection, such as replication of the VSV and AdV5.

### 3.5. Validation of Selected Antiviral Drugs with Pathogenic HTNV

In a first step of validation, we tested the compounds demonstrating high antiviral activity on VSV*ΔG-Luc(HTNV-G) in Vero E6 cells, as previously done with EIPA.

As observed in A549 cells, entry of VSV*ΔG-Luc(HTNV-G) into Vero E6 cells was also reduced in the presence of tetrandrine, monensin sodium salt and emetine dihydrochloride (Figure 6). No cytotoxicity was observed under the conditions of our assay. The overall similarity of the inhibitor profile in two different cell lines further supports a role for antiviral activity of tetrandrine, monensin sodium salt and emetine dihydrochloride on HTNV-G infection.

Next, we determined whether our compounds also inhibit infectious HTNV in BSL-3 laboratory, based on their efficacy to reduce HTNV-G pseudovirus infection in previous screens and characterizations. We selected three compounds, monensin sodium salt, emetine dihydrochloride, and tetrandrine, for follow up analysis.

Following 45 min drug pre-treatment, pathogenic HTNV was added on Vero E6 cells in the presence of the antiviral compound for 1 h. Cells were washed once and incubated with medium for 2 days. Cells were lysed and the level of virus infection was detected by RT-qPCR. Tetrandrine and emetine dihydrochloride drastically reduced HTNV infection (Figure 7A,B). Unfortunately, the treatment with monensin sodium salt showed cytotoxicity under the conditions of our assay. We then calculated the toxicity index (TI) to determine the relative safety of the drugs EIPA, emetine dihydrochloride and tetrandrine (Figure 7D). To this end, Vero E6 cells were treated under the infection assay conditions from 1 to 100 µM. Then, 48 h post-treatment, cells were lysed with 1% Triton-100 to obtain the background values. EIPA has a low toxicity effect compared to emetine dihydrochloride and tetrandrine (Figure 7C,D, toxic dose (TD_50_)). The relative safety of the compounds according to their respective inhibitory effect on HTNV infection indicated that emetine dihydrochloride is the compound showing the highest relative safety with a therapeutic index of >7 × 10^7^:1, while EIPA and tetrandrine have a lower therapeutic index of 729:1 and 6.92:1, respectively. Taken together, our data show that the screening platform is able to identify entry inhibitors.

## 4. Discussion

The lack of licensed antivirals to prevent VHF infection underlines the importance to develop methods for the identification of new therapeutics against these viruses. One promising line of research is the repurposing of drugs, which is time saving and therefore constitutes a major advantage in an outbreak context. Moreover, investigating the antiviral potential of natural compounds also represents a great advantage, as over 80% of the world’s population depends on traditional medicine, including the use of herbal medicinal products, as a first source of healthcare [11]. Moreover, many modern drugs are developed on active compounds isolated from plants. The use of medicinal plants to fight viral infections deserves further investigation.

To discover novel antiviral agents against viral entry into host cells, we used a powerful platform based on a recombinant VSV-pseudoviruses carrying an EGFP reporter gene [5,8,9]. This pseudovirus platform was used in multiple studies to uncover entry factors for hantaviruses, arenaviruses and filoviruses, among others. The studies showed a very close correlation of VSV-pseudoviruses with the authentic viruses [7,22,23,24,25,26,27,28]. In a first step, we demonstrated the potential of the system to evaluate antiviral compounds at a lower level of biological containment. This pseudotype platform can be applied for a broad range of enveloped viruses and thus provides great possibilities to identify drugs capable of blocking viral entry. The GFP reporter gene encoded in our pseudoviruses could be used as an alternative readout and demonstrated a very stable fluorescence.

Entry into host cells represents a crucial step of viral infection, which determines cell tropism and disease potential. To validate our screening platform, we used a potent inhibitor of macropinocytosis, EIPA, which showed antiviral activity against both HTNV-G and ANDV-G in our previous studies [5,8,9]. First, our analysis using the ImageXpress^®^ Micro XLS Widefield High-Content Screening System (Molecular Devices) to visualize and quantify the number of EGFP-infected cells showed a decrease in HTNV-G infection, when we treated the cells with 50 µM EIPA. We then further validated the antiviral activity of EIPA using pathogenic HTNV in the BSL-3 laboratory. We measured a dose-dependent inhibition of HTNV replication, with no impact on viability under our experimental conditions. Our findings indicate that targeting the sodium hydrogen exchanger could be a novel strategy to combat pathogenic HTNV.

In order to identify novel antiviral compounds from natural sources, we screened a phytochemical library. We identified five positive hits from the screen, and conducted further assays to characterize and validate their impact on virus infection. Two natural compounds demonstrated virus-specific activity. Rotenone showed antiviral activity mainly against ANDV-G and LASV-G. It was previously demonstrated to suppress the growth of Newcastle disease and herpes simplex viruses [29], as well as disrupting replication of western equine encephalitis viruses [30]. Nitrarine dihydrochloride, which is prescribed as an antihypertensive, sedative, antispastic and spasmolytic, has not been previously described as an antiviral. However, in our screening, EBOV-G and ANDV-G entry was drastically reduced in the presence of nitrarine dihydrochloride. Upon receptor-mediated endocytosis, EBOV, LASV, HTNV and ANDV are delivered to acidified endosomal compartments, where low pH is required for membrane fusion [31,32,33,34]. In the case of EBOV, LASV and ANDV, the fusion occurs in the late endosomes/lysosomes, while HTNV fuses in the early endosomes [9]. Since nitrarine and rotenone mainly have antiviral activity against EBOV-G, ANDV-G and LASV-G, this suggests that these two compounds are likely to block viruses that fuse in the late endosomal compartment.

Three additional compounds, tetrandrine, monensin sodium salt and emetine dihydrochloride seem to have a broad-spectrum inhibitory activity. Even though hanta-, filo-, and arenaviruses are distantly related, they share common features in their lifecycle, that can be blocked by small molecules [35,36,37]. Tetrandrine, which is used as an antineoplastic, demonstrated potent antiviral activity against all pseudoviruses tested. Interestingly, tetrandrine was shown to inhibit EBOV infection in human monocyte-derived macrophages and showed therapeutic efficacy in mice [38]. The identification of the antiviral activity of tetrandrine using EBOV pseudoviruses confirmed the robustness of our screening platform. The inhibitory activity affects the endosomal transport of viral particles by specifically blocking L-type calcium channels. Monensin sodium salt, an antibacterial and antimalarial drug, can prevent endocytosis by inhibiting the fusion of Semliki Forest virus [39], as well as the transport of viral glycoproteins to the cell surface [40]. More interestingly, monensin was previously shown to inhibit HTNV replication [41]. Emetine dihydrochloride, an anti-amebic, is a well-known inhibitor of cellular protein synthesis [42,43], viral RNA synthesis of SARS-CoV-2, Dengue, Zika and Ebola virus, as well as blocking EBOV entry [43,44,45]. By investigating entry and post-entry steps of VHFs, we demonstrated that monensin sodium salt can block entry, most likely by inhibiting exit of endosomal compartment, whereas emetine dihydrochloride probably inhibits viral replication.

By validating EIPA, emetine dihydrochloride and tetrandrine with authentic pathogenic HTNV, we observed a drastic reduction in infection, confirming again the robustness of our assay. Moreover, the therapeutic index we obtained provides additional evidence that entry inhibitors could be therapeutic strategies to fight hantaviruses infection. Whereas we found a similar impact of EIPA and tetrandrine with the pseudotype virus or the pathogenic HTNV, emetine dihydrochloride was much more potent against the pathogenic HTNV. This increased antiviral effect is likely due to the experimental setting, as the infection was followed during 48 h in presence of the drugs. During this period, the viruses can replicate and propagate. However, a major limitation in evaluating novel therapeutic strategies against HTNV is the lack of an animal model that exhibits clinical signs of the disease. The exposure of Syrian hamster, ferret, rhesus macaques and common marmoset to HTNV leads to asymptomatic infection [46].

In sum, the screening platform we have developed allows the identification of antiviral compounds against viral hemorrhagic fever viruses. In our study, we have identified several compounds, among them tetrandrine and emetine dihydrochloride, that could be used in novel antiviral strategies against hantaviruses.

## Figures and Tables

**Figure 1 viruses-13-00685-f001:**
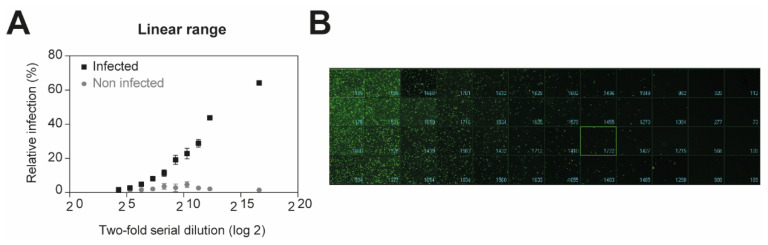
Determination of linear range for luminescent detection. Confluent monolayers of A549 cells were infected with two-fold serial dilution of VSV*ΔG-Luc(VSV-G) in DMEM 10% FCS for 16 h at 37 °C under 5% CO_2_. EGFP-positive infected cells were measured by high-content imaging. (**A**) Quantified representation of EGFP-expression. (**B**) Representative images of reporter virus-infected A549 cells. Images for two-fold serial dilutions are presented.

**Figure 2 viruses-13-00685-f002:**
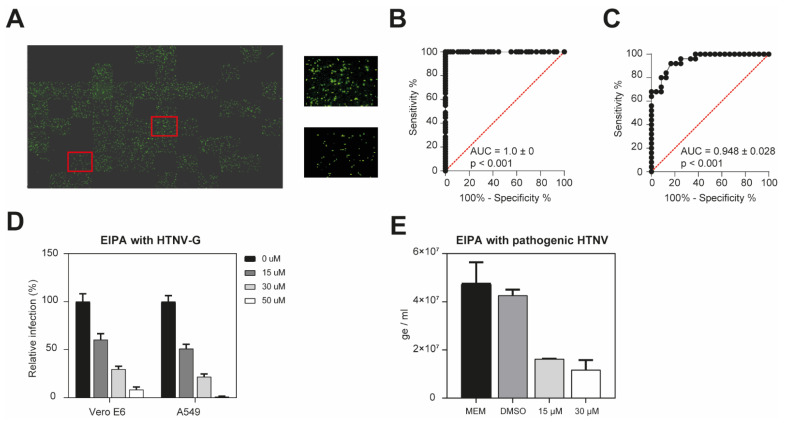
Performance of EGFP expression measurement as a method to differentiate non-treated from treated infected cells. (**A**) Representative image of a 96-well plate with two enlarged wells standing for treatment conditions. Three conditions were applied to confluent monolayers of A549 cells: no pre-treatment and no infection; no pre-treatment followed by infection with VSV*ΔG-Luc(HTNV-G) (upper right panel); or pre-treatment with 50 μM EIPA for 30 min (lower right panel), followed by infection with VSV*ΔG-Luc(HTNV-G) in the presence of the inhibitor. After 16 h of incubation, cells were washed, fixed with 2% PFA, and the nuclei were stained (DAPI). The number of infected cells was then analyzed through GFP expression using fluorescent microscope. Representation of EGFP expression on the XLS microscope. (**B**,**C**) ROC curves obtained for non-infected vs. infected cells (**B**) and for non-treated vs. EIPA-treated cells (**C**). The AUC and significance level are shown in each case. (**D**) EIPA inhibits HTNV-G in Vero E6 and A549 cells. Monolayers of Vero E6 or A549 cells were pre-treated with indicated concentration of EIPA for 45 min at 37 °C, followed by VSV*ΔG-Luc(HTNV-G) infection at 200 IU/well for 90 min. Cells were then washed and complete medium with 20 mM ammonium chloride was added and incubated for 16 h. The infection levels were assessed by counting the EGFP-positive infected cells. Data are means + SD (*n* = 3). (**E**) EIPA inhibits HTNV infection. Vero E6 cells were pre-incubated with EIPA at the indicated concentrations for 45 min at 37 °C, followed by HTNV infection for 1 h at 37 °C. Cells were subsequently washed once with medium and incubated with medium containing appropriate drug concentration for 48 h. Infection levels were assessed by RT-qPCR. Data are means + SD (*n* = 3) of genome per mL.

**Figure 3 viruses-13-00685-f003:**
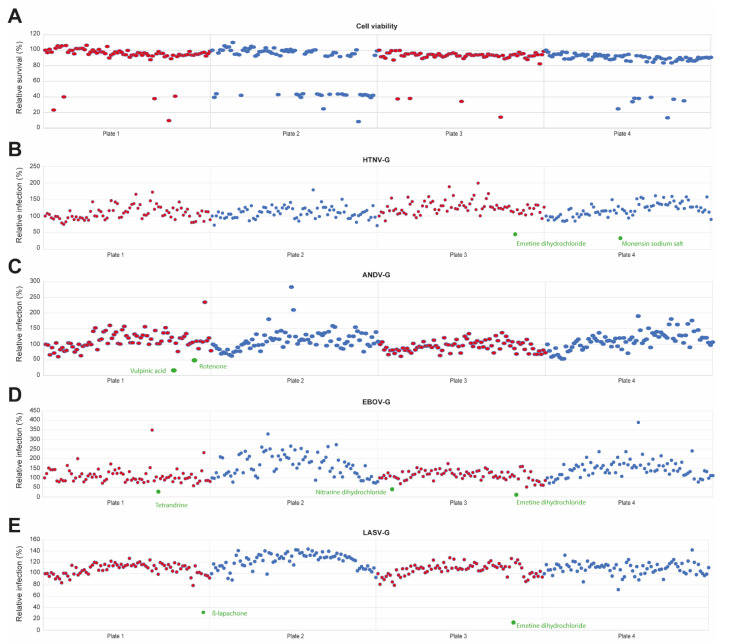
Screening of Prestwick Phytochemical library identified compounds reducing pseudoviruses entry. A549 cells were pre-treated with 10 µM of compounds for 45 min and infected with VSV*ΔG-Luc(HTNV-G), VSV*ΔG-Luc(ANDV-G), VSV*ΔG-Luc(EBOV-G) or VSV*ΔG-Luc(LASV-G) in presence of the drugs. After 2 h, cells were washed 3 times with medium containing 20 mM ammonium chloride, followed by 16 h of incubation in the presence of the lysosomotropic agent. Infection was detected by EGFP and the control (DMSO) set at 100%. (**A**) Cell viability was measured by the CellTiter-Glo^®^ assay. The compounds demonstrating a reduction of >20% in cellular ATP levels were considered toxic. (**B**–**E**) Infection was detected by EGFP expression and the control (DMSO) set at 100%. The compounds reducing viral entry by >50% are highlighted in green and names wrote underneath.

**Figure 4 viruses-13-00685-f004:**
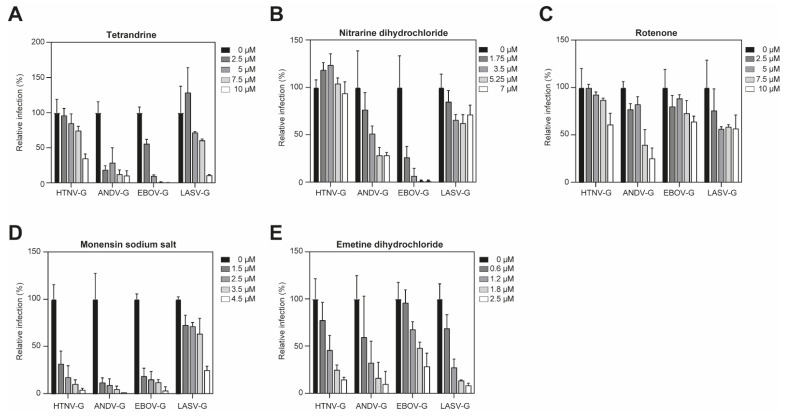
Tetrandrine (**A**), nitrarine dihydrochloride (**B**), rotenone (**C**), monensin sodium salt (**D**) and emetine dihydrochloride differentially (**E**) affect infection of VSV*ΔG-Luc(HTNV), VSV*ΔG-Luc(ANDV), VSV*ΔG-Luc(EBOV) and VSV*ΔG-Luc(LASV) pseudotypes in A549 cells. A549 cells were pre-treated with different compounds at increasing concentrations for 45 min at 37 °C, followed by infection with the pseudoviruses in the presence of the drug for 90 min at 200 IU/well. Complete medium with 20 mM ammonium chloride was added and incubated for 16 h. The infection levels were assessed by counting the EGFP-positive infected cells. Cell toxicity was assessed in parallel on uninfected cells by measuring ATP content. Data are means + SD (*n* = 3).

**Figure 5 viruses-13-00685-f005:**
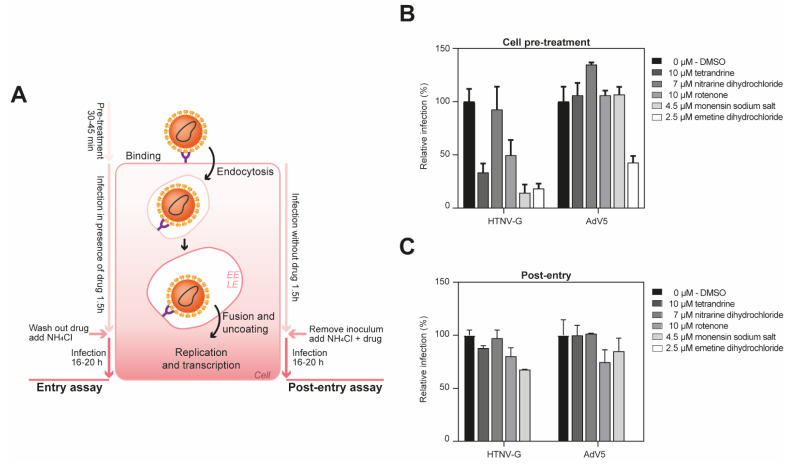
Emetine dihydrochloride is broad-spectrum inhibitor in A549 cells. (**A**) Schema of the entry (left) and post-entry assay (right). EE, early endosome; LE, late endosome. (**B**) A549 cells were pre-treated with indicated compounds at specific concentrations for 45 min at 37 °C, followed by infection with the pseudoviruses in the presence of the drug at 200 IU/well. After 90 min, complete medium with 20 mM ammonium chloride was added and further incubated for 16 h. The infection levels were assessed by counting the EGFP-positive infected cells. Cell toxicity was assessed in parallel on uninfected cells by measuring ATP content. Data are means + SD (*n* = 3). (**C**) Time-of-addition experiment. A549 cells were infected with VSV*ΔG-Luc(HTNV-G), or AdV5-GFP at 200 IU/well and treated with indicated drug concentration after 1.5 h of infection. Infection levels were assessed by counting EGFP-positive infected cells. Data are means + SD (*n* = 3).

**Figure 6 viruses-13-00685-f006:**
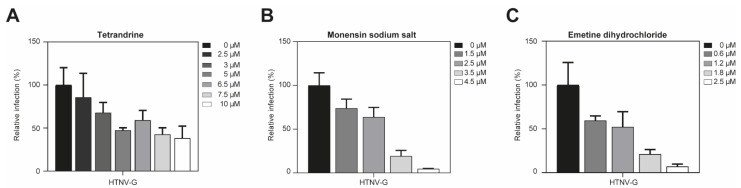
Treatment with tetrandrine, monensin sodium salt and emetine dihydrochloride in Vero E6 cells. (**A**–**C**) Monolayers of Vero E6 cells were pre-treated with the indicated drugs at given concentrations and then infected with VSV*ΔG-Luc(HTNV-G). After 90 min, cells were washed and complete medium with 20 mM ammonium chloride was added and further incubated for 16 h. The infection levels were assessed by counting the EGFP-positive infected cells. Data are means + SD (*n* = 3).

**Figure 7 viruses-13-00685-f007:**
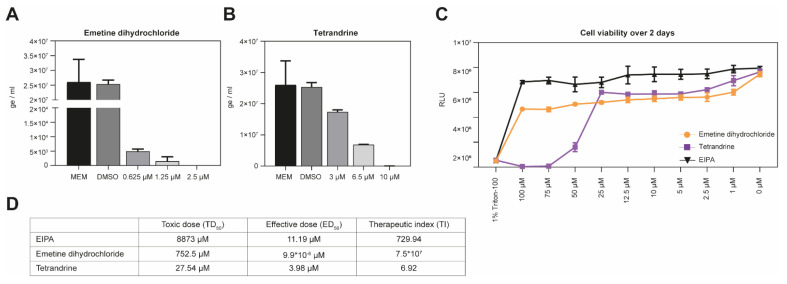
Validation with authentic pathogenic viruses. (**A**,**B**) Vero E6 cells were pre-incubated with emetine dihydrochloride (panel A), or tetrandrine (panel B) at the indicated concentrations for 45 min at 37 °C, followed by HTNV infection for 1 h at 37 °C. Cells were subsequently washed once with medium and incubated with medium containing appropriate drug concentrations for 48 h. Infection levels were assessed by RT-qPCR. Data are means + SD (*n* = 3) of genome per ml. (**C**) Cell viability over two days. Monolayers of Vero E6 cells were treated with the indicated concentration of emetine dihydrochloride or tetrandrine under the assay conditions (**A**,**B**). After 48 h, intracellular ATP levels were measured using CellTiter-Glo^®^ assay. Data are means ± SD (*n* = 3) of relative light units (RLU). (**D**) Calculation of the therapeutic index (TI). The therapeutic index is defined as toxic dose (TD_50_)/effective dose (ED_50_).

## Data Availability

All data are available under request.

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
