# Peer review of "Identification of Novel Antiviral Compounds Targeting Entry of Hantaviruses"

_viruses, 2021, doi:10.3390/v13040685_

Round 1

Reviewer 1 Report

Mayor, et al present the results of a study to evaluate the viral entry blocking potential of a library of natural compounds. The manuscript is well-written and my feedback minor.

  1. Figure 3 is very busy. Is it possible to further identify or make it easier for the reader to pick out the dots (and perhaps label with the compound name) the compounds that meet criteria?
  2. There is no mention of advancing promising antivirals into animal models. Are there plans to do this?

Reviewer 2 Report

Nowadays, the need for antiviral drugs is obvious and screening of drug libraries in cell culture experiments may be the first step in the identification of potent candidates. Therefore, the work presented by Mayor and colleagues is of special interest. However, the design of the study demonstrates some weakness and presentation of data – especially in the last figure - raises several questions. Major points of concern are the use of two different cell lines and the confusing calculation of TD50, ED50 and TI. Authors should give reasons for the use of A549 and Vero E6 cells and carefully revise figure 6 to demonstrate the suitability of the pseudotype platform and to confirm the antiviral activity of emetine.

Major points:

abstract/introduction: The text passages start with hantaviruses, but several other viral hemorrhagic fever viruses were tested. Therefore, the abstract, introduction should be not so focused on hantaviruses. In addition, the novel taxonomy of hantaviruses: Hantaviridae and orthohantaviruses should be mentioned.

Line 11: “lack of licensed vaccines”: Hantavax is available for use in several Asian countries, but has not been approved for use in Europe or U.S.A.

Line 114: The number of cells inoculated with infectious units should be included (MOI).

Line 220, 224, figure 2D: Why is a concentration of 50 µM of EIPA used in A549 cells and a concentration of 30µM in Vero E6 cells? A decrease of about 65% in viral production is observed in Vero E6 cells. What is the decrease in A549 cells? In this figure a comparison between inhibition of pseudotype and pathogenic virus is necessary to show that results are comparable; otherwise the method is not suitable for screening.

Line 205: “A549 cells are susceptible to HTNV infection.” Why do authors use Vero E6 cells in infection studies, if A549 cells are susceptible to HTNV infection? The results would be comparable by using one cell type.

Figure 3A shows viability, but the Y-axis is labeled “Relative infection”?

Figure 3B-E: it would be of interest to identify the dots for the five compounds in the four plots. Authors should indicate the dots by color or arrows in the figure.

Figure 4: Figure 4A: SD is missing for LASV incubated with 5, 7.5 and 10 µM tetrandrine. Figure 4B: Why is there no inhibition of HTNV by nitrarine dihydrochloride? Figure 4C: Why is there no >50% inhibition for HTNV, EBOV and LASV for rotenone? In the initial screening these compounds were selected due to a reduction of viral entry by >50% (line 264). Was the inhibition lower in the screening or was it not observed for all viruses? Why was a concentration of 7 µM of nitrarine used in figure 4B instead of 10 µM used in figure 3?

Figure 5B: The inhibition shown in figure 5B is much lower than in figure 4. Relative infection with psHTNV is much higher in cells incubated with tetrandrine and monensin as shown for figure 4.

Figure 6 is really confusing. 6A and B show the infection levels quantified by qPCR. Two concentrations were shown for emetine and three for tetrandrine. From figure 6b, an estimated ED50 value of 3.98 µM for tetrandrine as stated in 6D is plausible. However, the ED50 for emetine is not to clear. Authors should show a range of concentrations for emetine that covers the ED50 value. The same is true for the calculation of TD50. Figure 6C does not provide concentrations with 50% viability, it is not clear, how the TD50 is calculated from these experiments. EIPA data are completely missing.

Figure 6 should validate the results from assays with pseudotyped viruses in the context of infection with pathogenic wild type virus. However, the range of drug concentration and calculated values in the infection studies do not correspond to results from experiments with pseudotyped HTNV. From figure 4A the ED50 of tetrandrine is between 7.5 and 10 µM and from figure 4E the ED50 of emetine is between 0.6 and 1.2 µM for HTNV-G. In wild type infection the ED50 is 3.96 and 9.9*10^-6 for tetrandrine and emetine, respectively. The ED50 values between pseudotype infection and wildtype infection are not comparable. Different cell types (A549 versus Vero E6 cells) were used and also the methods for quantification were different (number of infected cells versus qPCR). These differences may influence the results. However, if results from pseudotype assay are so different from infection studies with Hantaan virus, the method is not suitable for the identification of antiviral drugs.

Discussion: the work of Schmaljohn et al. (1986) using monensin as hantavirus inhibitor should be cited.

Minor points:

Line 144: 48 H

Line 310: virus "who"

Experiments using pseudotyped viruses should be labeled as HTNV-G or psHTNV to be distinguishable from experiments with pathogenic wildtype viruses.
